# Evaluation of Dynamic Contrast-Enhanced and Oxygen-Enhanced Functional Lung Magnetic Resonance Imaging in Chronic Obstructive Pulmonary Disease Patients

**DOI:** 10.3390/diagnostics13233511

**Published:** 2023-11-23

**Authors:** Rohit K. Srinivas, Mandeep Garg, Uma Debi, Nidhi Prabhakar, Sahajal Dhooria, Ritesh Agarwal, Ashutosh Nath Aggarwal, Manavjit Singh Sandhu

**Affiliations:** 1Department of Radio Diagnosis and Imaging, Post Graduate Institute of Medical Education and Research, Chandigarh 160012, India; rohitksrinivas@gmail.com (R.K.S.); debi_uma@yahoo.co.in (U.D.); nidhirajpalprabhakar@gmail.com (N.P.); manavjitsandhu@yahoo.com (M.S.S.); 2Department of Pulmonary Medicine, Post Graduate Institute of Medical Education and Research, Chandigarh 160012, Indiaagarwal.ritesh@outlook.in (R.A.); aggarwal.ashutosh@outlook.com (A.N.A.)

**Keywords:** MRI, lung MRI, functional lung MRI, oxygen enhanced, dynamic contrast enhanced, ventilation/perfusion, chronic obstructive pulmonary disease

## Abstract

Chronic obstructive pulmonary disease (COPD) is a chronic respiratory condition characterized by obstruction of airways and emphysematous lung tissue damage, with associated hypoxic vasoconstriction in the affected lung parenchyma. In our study, we evaluate the role of oxygen-enhanced (OE) MRI and dynamic contrast enhanced (DCE)-MRI in COPD patients for assessment of ventilation and perfusion defects and compared their severity with clinical severity. A total of 60 patients with COPD (diagnosed based on clinical and spirometry findings) and 2 controls with normal spirometry and no history of COPD were enrolled. All patients underwent MRI within 1 month of spirometry. OE-MRI was performed by administering oxygen at 12 L/min for 4 min to look for ventilation defects. DCE-MRI was performed by injecting intravenous gadolinium contrast, and perfusion abnormalities were detected by subtracting the non-enhanced areas from the first pass perfusion contrast images. A total of 87% of the subjects demonstrated ventilation and perfusion abnormalities on MRI independently. The lobe-wise distribution of ventilation and perfusion abnormalities correlated well with each other and was statistically significant in all lobes (*p* < 0.05). The severity of ventilation-perfusion defects also correlated well with clinical severity, as their median value (calculated using a Likert rating scale) was significantly lower in patients in the Global initiative for chronic Obstructive Lung Disease (GOLD) I/II group (3.25) compared to the GOLD III/IV group (7.25). OE- and DCE-MRI provide functional information about ventilation-perfusion defects and their regional distribution, which correlates well with clinical severity in patients with COPD.

## 1. Introduction

Chronic obstructive pulmonary disease (COPD) is a heterogeneous lung condition characterized by chronic respiratory symptoms (dyspnea, cough, sputum production, and/or exacerbations) due to abnormalities of the airways (bronchitis, bronchiolitis) and/or alveoli (emphysema) that cause persistent, often progressive, airflow obstruction [1,2]. There can be many causative factors for COPD, like genetics, developmental factors, asthma, or infections, but persistent exposure to noxious particles is the most common cause, with cigarette smoking being the biggest culprit. Gas exchange at the alveolar level in normal healthy lungs is maintained through optimal ventilation and perfusion ratios. In COPD, the ventilation component is hampered due to lung parenchymal destruction and/or airway obstruction. There is associated hypoxic vasoconstriction in the areas with impaired ventilation, leading to reduced perfusion of that area and shunting to the well-ventilated areas of the lungs. The lobar distribution of these perfusion abnormalities may not necessarily match the structural alterations [3,4].

Spirometry has traditionally been used as a reliable and objective measurement tool for the diagnosis and stratification of COPD. A post-bronchodilator forced expiratory volume in the first second (FEV_1_)/forced vital capacity (FVC) ratio of <0.7 is the cut-off value for airflow limitation used to diagnose COPD. FEV_1_ (% predicted) determines the severity of the disease and is classified into four groups using the Global initiative for chronic Obstructive Lung Disease (GOLD) criteria, with an FEV_1_ ≥ 80% (predicted) as GOLD I (mild), 50–79% (predicted) as GOLD II (moderate), 30–49% (predicted) as GOLD III (severe), and <30% (predicted) as GOLD IV (very severe) [1]. The additional testing parameters include assessment of lung volume, arterial blood gas measurement, oximetry, diffusion capacity, cardiopulmonary exercise testing, and assessment of physical activity [5]. However, these common diagnostic modalities give information only about the overall lung function status and do not give information about the structural alterations or regional distribution of the disease [6]. 

Currently, computed tomography (CT) of the chest is being used widely for imaging in patients with COPD. However, CT does not provide functional information about the lungs and also carries an increased risk of radiation-induced health effects as these patients often need multiple, repeat imaging to monitor disease progression/response to therapy [7]. This has prompted researchers to explore other non-ionizing imaging modalities like magnetic resonance imaging (MRI) [8]. Traditionally, the use of MRI in lungs had been limited due to the intrinsic architecture of lungs, air-filled spaces, and respiratory and cardiac motion. These resulted in a low signal-to-noise ratio (SNR) and decreased image quality. However, MR technology has undergone a wide change in the past few years with the advent and implementation of intravenous and inhalational MR contrast. This has resulted in lung MRI gaining increased attention in the evaluation of various thoracic disorders [9]. Hyperpolarized noble gases like helium (^3^He) and xenon (^129^Xe) have been used as inhalational agents for the evaluation of lung microstructure and to demonstrate ventilation/perfusion (V/Q) defects [10]. Phase-resolved functional lung MRI without the use of intravenous contrast agent has also shown good repeatability for V/Q scans [11]. 

Oxygen-enhanced (OE)-MRI utilizes the paramagnetic effect of oxygen (O_2_) to help in identifying areas with ventilation defects [12]. After oxygen inhalation, the dissolved oxygen in the blood pool and tissue water causes T1 shortening of normally ventilated lungs, which is seen as an increase in signal intensity when compared to a T1-weighted image in room air (RA). This T1 shortening is reduced in COPD patients and has been seen to correlate with pulmonary function test (PFT) results and pulmonary diffusion capacity [13]. 

Dynamic contrast-enhanced (DCE)-MRI demonstrates areas of hypoxic vasoconstriction noted in regions of the lung with impaired ventilation. MR perfusion images are derived from subtracting the non-enhanced areas from the first-pass perfusion contrast images. The diagnostic accuracy of MR perfusion is high in identifying perfusion defects of the lung [9]. There is a significant reduction in the blood vessels per unit volume in the COPD-affected lung due to increased distal air space from emphysema, hyperinflation, and hypoxic vasoconstriction; DCE-MRI visualizes these pathologies indirectly [14]. We used a combination of OE-MRI and DCE-MRI in the current study to detect ventilation-perfusion defects in COPD-affected lungs and to compare the results with the clinical severity of the disease. 

## 2. Materials and Methods

The Institutional Review Board gave its approval for this study. Written, informed consent for examination and data evaluation was obtained from all subjects. The work was carried out in accordance with The Code of Ethics of the World Medical Association (Declaration of Helsinki).

This prospective observational single-center study was carried out between December 2018 to March 2020. A total of 60 patients with COPD (diagnosed based on clinical and spirometry findings) who presented to our hospital during the study period were enrolled. Inclusion criteria for the participants were adults aged greater than 18 years of age with a confirmed diagnosis of COPD (FEV1/FVC < 0.7). Exclusion criteria were pregnancy, age <18 years, patients on chronic oxygen therapy, contraindication to intravenous contrast, and contraindication to MRI (patients with claustrophobia, magnetic implants, pacemaker, etc.). Two controls were also registered in the study who had normal spirometry values and no prior history of COPD. A flowchart depicting the study design is shown in Figure 1.

Spirometry was performed in all patients using an Easy connect intelligent workflow spirometer (New Diagnostic Design Medical Technologies Inc., Andover, MA, USA). The study participants who met the inclusion criteria underwent thoracic MRI within 1 month of undergoing the spirometry test. MRI was performed using a 1.5 Tesla MR unit (MAGNETOM AERA; Siemens Medical Solutions, Malvern, PA, USA) equipped with a body coil and high-performance gradient system. MRI was acquired from lung apices to the domes of the diaphragm with an Axial T2W Half-Fourier Acquisition Single-Shot Turbo Spin-Echo (HASTE), a T2W True Fast Imaging with Steady-State Free Precession (TRUFI), a T2W BLADE (the proprietary name for periodically rotated overlapping parallel lines with enhanced reconstruction), and diffusion-weighted imaging (DWI). T1-weighted images in room air (RA) were captured using the inversion recovery (IR) snapshot fast low angle shot (FLASH) sequence in coronal plane. Oxygen was administered at 12 L per minute (L/min) for 4 min, and the sequence was repeated to obtain T1-weighted images (OE). After oxygen administration, a state of pulmonary hyperoxia usually occurs in about 3 min in normal healthy lungs, resulting in a reduction in T1 relaxation and an increase in T1 signal intensity (SI), which is about 8–10% compared with that of the T1 signal intensity at RA. Therefore, an increase in the SI values on T1 (OE) when compared to T1 (RA) was considered normal, and if there was no change/decrease, the result was labelled as affected. T1 (RA) and T1 (OE) were later translated into red-green-blue color-coded maps ranging from 1 ms (dark blue) to 1800 ms (dark red). Areas of green in a background of blue lung parenchyma was considered normal, and a reduction in the green foci was considered affected. Time-resolved angiography with stochastic trajectories (TWIST) ANGIO 3D sequences was performed for perfusion images after injecting intravenous gadolinium contrast (0.05 mmol/kg body weight) at a rate of 5 mL/s, which was followed with a saline chase. Perfusion images in grey scale were obtained by subtracting the non-contrast images from the point of highest enhancement in lung tissue. Contrast-related enhancement of the segmental and subsegmental pulmonary vessels was considered normal. Reduced caliber or non-enhancement of these vessels in each lobe was considered affected. The mean time taken for the completion of MRI scans was 14.67 ± 0.397 min. Breath-hold was only required for T1-weighted sequences for a mean duration of 18.6 ± 0.472 s. The detailed information about the MRI protocol used in this study is summarized in Table 1. 

Two experienced thoracic radiologists with 24 years and 12 years of experience, respectively, evaluated the MRI, who were blinded from the clinical data and spirometry details. Kappa value of agreement between the readers was ~0.8 for this study. For image analysis (OE MRI and DCE MRI), the right lung was divided into the right upper lobe (RUL), right middle lobe (RML), and right lower lobe (RLL), and the left lung was divided into the left upper lobe (LUL) and left lower lobe (LLL). The division of lungs into different lobes was performed manually, and no automation/software was used. The severity of perfusion defects in each lung zone was rated with a semi-quantitative 3-point Likert rating scale, where 0 = normal, 1 = <50% of lung zone affected, and 2 = >50% of lung zone affected. The T1 image after oxygen administration was compared with the T1 image in room air based on signal intensity values. A region of interest (ROI) circle with a minimum diameter of 1 cm was drawn in the affected region excluding the bronchi and vessels on the T1 image (RA) and the corresponding location on the T1 image (OE). The mean SI values were compared. Later, color-coded maps were generated from T1 OE, and a semi-quantitative 3-point Likert rating scale with 0 = normal, 1 = <50% of lung zone affected, and 2 = >50% of lung zone affected was used for their evaluation. The quality of MR images obtained from each sequence was assessed as follows: grade I—severe artefacts (images not of diagnostic value), grade II—images impaired by artefacts but are of diagnostic value, and grade III—no artefacts. Ancillary MR findings such as areas of consolidation, pleural effusion, lymph nodes, etc., were also noted. Methods for OE and DCE MRI data analysis were similar to those reported by Jobst et al. [12]. 

SPSS 22 version software was used for statistical analysis. The categorical and continuous data are presented in the form of frequencies/proportions and mean/standard deviation, respectively. The tests of significance for the mean difference between more than two groups (quantitative and qualitative data) were ANOVA (analysis of variance) and the Kruskal–Wallis test, respectively. Spearman’s correlation coefficient (r) was calculated to find the correlation between two qualitative variables. *p*-value (probability of the test results being true) of <0.05 was considered statistically significant after assuming all the rules of statistical tests. 

## 3. Results

The demographics and main functional data of study participants are summarized in Table 2. In this study cohort, 93.3% of the subjects were men and 6.7% were women, with a mean age of 61.73 years. Shortness of breath was the predominant symptom (96.7%), followed by cough (63.3%). According to the COPD severity GOLD classification, 6.7% patients were Grade I, 33.3% were Grade II, 53.3% were Grade III, and 6.7% were Grade IV. FEV_1_ (% predicted) was <50 in 60% of the patients, while it was >50 in the other 40%.

Ventilation and perfusion defects were seen equally in 52/60 patients each. Lobar analyses of the perfusion and ventilation abnormalities were assessed independently for all the patients. Ventilation defects were identified in 202 of 300 lobes (Figure 2), while perfusion defects were identified in 214 of 300 lobes (Figure 3). Lobar distributions of ventilation and perfusion defects were statistically significantly different, with a positive correlation in all lobes (RUL: *p* < 0.001, r = 0.618; RML: *p* = 0.002, r = 0.532; RLL: *p* < 0.001, r = 0.652; LUL: *p* = 0.009, r = 0.474; LLL: *p* = 0.001, r = 0.696).

The sum of the Likert values obtained from the lobar distribution of ventilation was calculated. Similarly, these values were calculated for the perfusion images. An average of the two and the median values were then calculated and compared in two groups—GOLD I/II (mild COPD) and GOLD III/IV (severe COPD) (Figure 4). The median value in the GOLD I/II group (3.25) was significantly lower than that in the GOLD III/IV group (7.25). We also compared the means of individual groups of GOLD criteria. There was a gradual increase in the mean values obtained from the GOLD I to III groups, while it showed decrease in GOLD IV group (Table 3). However, these were found to be statistically non-significant. 

Ancillary findings of consolidation and nodules (6/60), fibroatelectatic and fibrobronchiectatic changes (10/60), goiter with mediastinal extension and significant tracheal luminal compromise (1/60), isolated RML collapse (1/60), and aneurysmal dilatation of the infrarenal aorta (1/60) were also noted. 

All the sequences were of diagnostic image quality, with none having grade I image quality. The grade II image quality was seen in T2 HASTE (56.6%), T2 BLADE (76.6%), and DWI (96.7%) sequences, while grade III image quality was found in T2 TRUFI (56.7%), T1 (80%), and TWIST Angio (93.3%) sequences (Figure 5 and Figure 6).

## 4. Discussion

COPD is the most common chronic respiratory condition in the world, with a global prevalence of ~13.1%, and is associated with high morbidity and significant socio-economic burden [15]. Spirometry and CT scanning currently serve as the accepted clinical and imaging evaluation modalities. The risk of ionizing radiation associated with repeated CT imaging and its inability to provide functional information have led to the exploration of lung MRI as an alternative imaging modality [16]. Conventionally, the low SNR in the lung parenchyma limited the utility of lung MRI due to low proton density and numerous air tissue interfaces leading to high local susceptibility [17]. However, the recent technological advancements in MRI sequences, use of inhalational gases, DCE-MRI, ultrashort echo time (UTE), multi-coil parallel imaging, zero echo time MRI, and phase-resolved MRI have expanded the scope of MRI in lung diseases [18]. 

There is decreased T1 shortening noted after the administration of oxygen in COPD patients due to hypoxic vasoconstriction and emphysematous tissue loss, and this principle is used in OE-MRI to differentiate areas of impaired ventilation from normally ventilated lung [19]. DCE-MRI helps with the detection of decreased pulmonary microvascular blood flow in COPD, which is seen even in mild disease and in areas of the lungs without obvious emphysematous changes [20]. Our study investigated the role of OE-MRI in the detection of ventilation defects and DCE-MRI in the detection of perfusion abnormalities and correlated their lobar distribution. The severity of the ventilation-perfusion defects was also correlated with that of clinical severity (GOLD I–IV). 

In the present study, ~87% of patients and ~67.33% of lobes demonstrated ventilation defects, while ~87% of patients and ~71.33% of lobes demonstrated perfusion defects. The morphologic comparison of the lobar distributions of ventilation and perfusion defects was statistically significant in all lobes and showed a positive correlation (RUL: *p* < 0.001, r = 0.618; RML: *p* = 0.002, r = 0.532; RLL: *p* < 0.001, r = 0.652; LUL: *p* = 0.009, r = 0.474; LLL: *p* = 0.001, r = 0.696). Our findings are in agreement with the results of a previous study conducted by Jobst et al. [12] with a sample size of 20 participants (while our study group was relatively large with 60 patients), which concluded that the areas of ventilation defects corresponded to the areas of perfusion defects. 

In our patient cohort, the median of Likert score values obtained from the lobar distributions of ventilation and perfusion were significantly lower in the GOLD I/II (3.25) group than in the GOLD III/IV (7.25) group. These findings are also in concordance with the study conducted by Jobst et al., in which the authors observed that abnormal T1 values correlated with results of the lung function tests [12]. Our findings are also in sync with another study conducted by Jang YM et al., which mentions that MR perfusion defects correlated with the worsening of PFT parameters [21]. Our study also compared the mean scores of individual groups of GOLDs criteria; however, the results were statistically non-significant. There was a gradual increase in the mean value obtained from GOLD groups I to III, with a decrease in the value in the GOLD IV group. In GOLD IV cases, the severity of perfusion abnormalities was far higher than that of the ventilation defects, and we theorize that this is possibly related to the greater slice thickness of the T1 maps (15 mm) when compared to the perfusion images (5 mm). The areas of residual normal blood contents within the MR slice might have outweighed the areas of decreased blood fraction, since the measured T1 is an image weighted by proton density [12].

The mean time taken for the acquisition of MRI images using our study protocol was 14.67 ± 0.397 min and breath-hold was needed only for T1 sequences. This was nearly half the time than that reported by Jobst et al. [12], which was about 30 min of scan time for every patient. It is imperative to note the ancillary findings such as consolidation and nodules, as they suggest a probable underlying cause for acute exacerbation [22]. Six patients among our study group showed the presence of consolidation and nodules, suggesting an underlying infective etiology. In developing countries like India, with a high burden of tuberculosis (TB), fibroatelectatic and fibrobronchiectatic changes (seen in ten patients in our cohort) could indicate a past history of TB. This is an independent risk factor for COPD, and the same has also been reported in another study by Sarkar M et al. [23]. Caution also needs to be exercised to differentiate and diagnose cases of reactivation tuberculosis, which could be the cause of acute exacerbation. One patient in our cohort also showed an enlarged thyroid gland with mediastinal extension and significant tracheal luminal compromise, which could have been the cause of obstructive symptoms in this patient. 

### Limitations

There were a few limitations in our study. First, it has a small sample size of 60 patients and a small representation (6.7%) of the female sex. Second, few controls were used due to ethical constraints. Third, the morphological information obtained was not compared with that of CT scan, which is currently the imaging modality of choice. Our study also did not compare the MRI findings with those of other established methods of lung perfusion detection like V/Q scanning and multi-energy CT. And lastly, we included only stable patients with COPD and could not evaluate the efficacy of OE- and DCE-MRI in sick/intubated patients requiring chronic oxygen support. 

## 5. Conclusions

OE- and DCE-MRI provided morphological as well as functional information about ventilation and perfusion defects in COPD, which correlated well with the clinical severity of COPD. Thus, MRI can be used as an adjunct to spirometry and has the potential to act as a good alternative to CT as the first choice for imaging modality. The fast acquisition time and no additional hardware/software support make OE- and DCE-MRI a suitable choice for COPD patients. However, in view of the high incidence of the disease in the population, more prospective and multicentric studies with a larger sample size are required to validate its efficacy. 

Summary Statement: 

Oxygen-enhanced and dynamic contrast-enhanced MRI provides morphological and functional information about ventilation and perfusion defects in chronic obstructive pulmonary disease patients and correlates well with clinical severity.

Key Results:

The lobar distribution of ventilation abnormalities in chronic obstructive pulmonary disease patients correlates with that of the perfusion abnormalities. There was a significant difference in the ventilation-perfusion abnormalities between Global initiative for Obstructive Lung Disease (GOLD) I/II criteria group versus GOLD III/IV group.

## Figures and Tables

**Figure 1 diagnostics-13-03511-f001:**
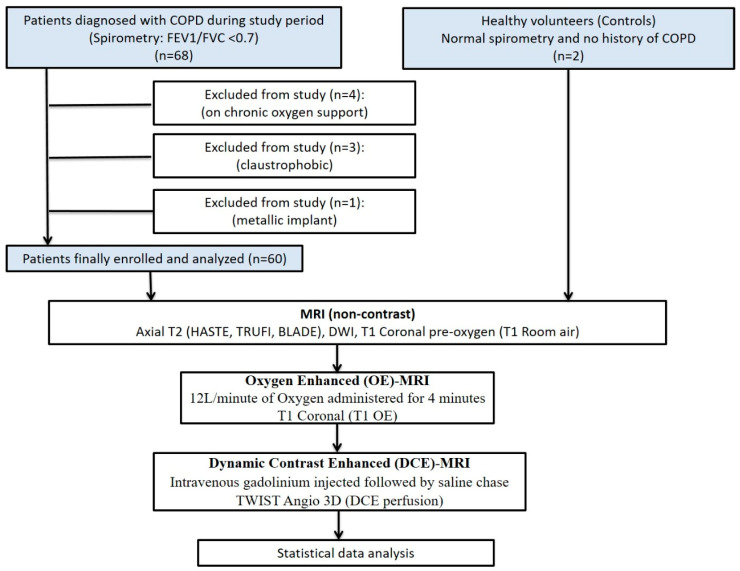
Flow chart depicting the study design along with inclusion and exclusion criteria.

**Figure 2 diagnostics-13-03511-f002:**
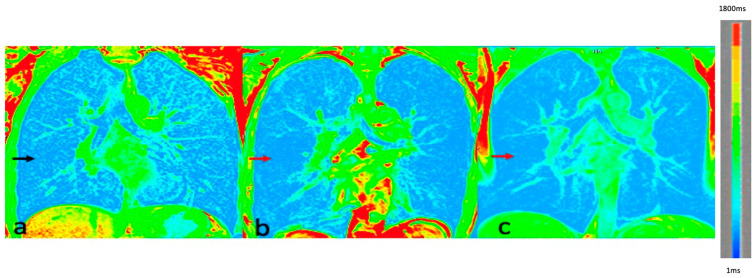
Color-coded T1 oxygen-enhanced (OE)-MRI images: (**a**) normal ventilation map (adequate green foci in the background of blue lung parenchyma—black arrow) in a 55-year-old male patient with GOLD I severity score, (**b**) mild ventilation defects in bilateral lungs (mild reduction in green foci in the background of blue lung parenchyma—red arrow) in a 61-year-old male patient with GOLD III severity score, (**c**) severe ventilation defects in bilateral lungs (minimal green foci in the background of blue lung parenchyma—red arrow) in a 55-year-old male patient with GOLD IV severity score. (Please note: artefacts noted in the periphery of lungs due to probable FOV parameters during acquisition).

**Figure 3 diagnostics-13-03511-f003:**
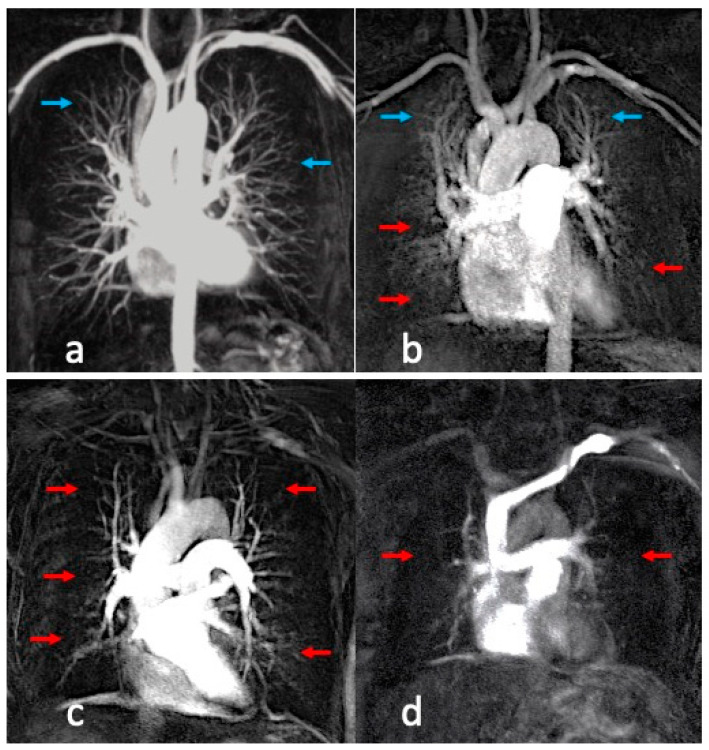
Dynamic contrast-enhanced (DCE)-MRI images demonstrating perfusion abnormalities: (**a**) normal perfusion map (well enhanced segmental and subsegmental pulmonary vessels—blue arrows) in a 42-year-old healthy volunteer/control; (**b**) mild perfusion defects (well-enhanced segmental vessels with poorly enhanced subsegmental vessels) in right middle lobe, right lower lobe, and left lower lobe (red arrows), with normal perfusion map in bilateral upper lobes (blue arrows) in a 65-year-old female patient with GOLD II severity score; (**c**) mild perfusion defects (well-enhanced segmental vessels with poorly enhanced subsegmental vessels) in right upper lobe, right middle lobe, right lower lobe, left upper lobe, and left lower lobe (red arrows) in a 69-year-old male patient with GOLD III severity score; (**d**) severe perfusion defects (poorly enhanced segmental and subsegmental pulmonary vessels) in right upper lobe, right middle lobe, right lower lobe, left upper lobe, and left lower lobe (red arrows) in a 62-year-old male patient with GOLD IV severity score.

**Figure 4 diagnostics-13-03511-f004:**
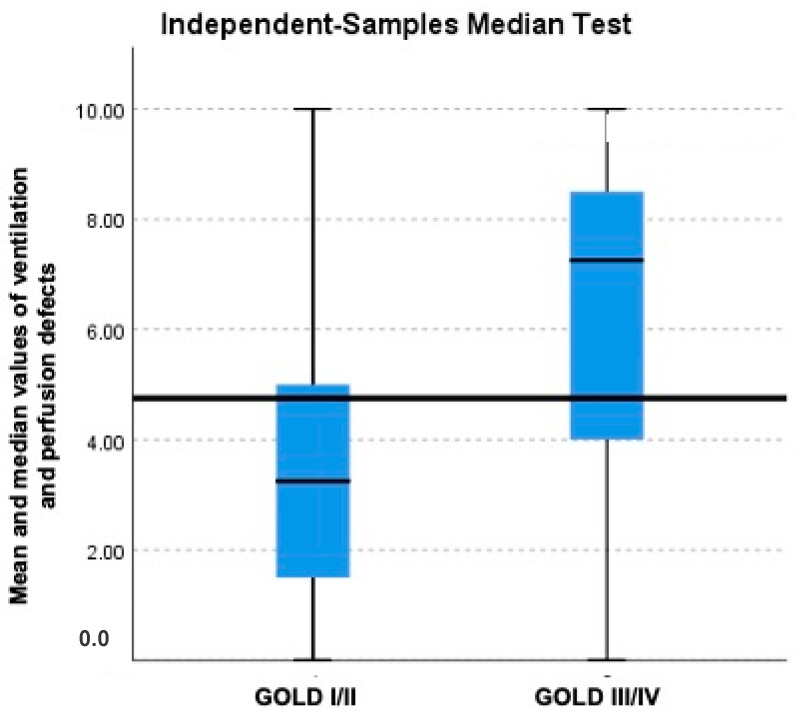
Box and whisker plot comparing the mean (blue box) and median (black line) values of ventilation-perfusion defects with GOLD I/II and GOLD III/IV severity score.

**Figure 5 diagnostics-13-03511-f005:**
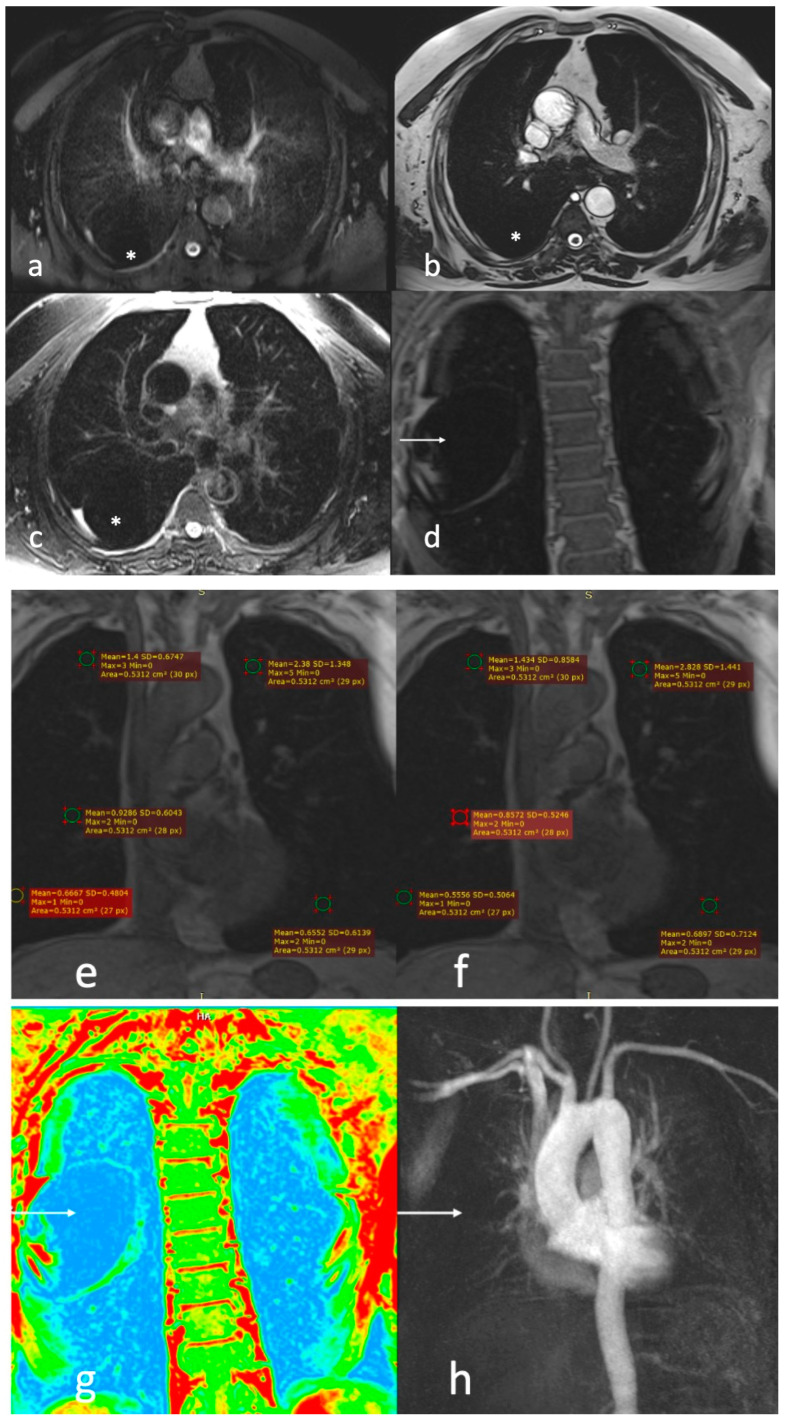
A 63-year-old male patient with GOLD III severity score: (**a**) T2 HASTE, (**b**) T2 TRUFI, and (**c**) T2 BLADE images showing bilateral hyperinflated lungs with (*) paraseptal emphysematous changes; (**d**) T1 image in room air (RA) showing a large bulla in the right upper lobe (white arrow); (**e**) T1 image in RA; and (**f**) T1 image, oxygen-enhanced (OE), showing ventilation defects in the right upper lobe, right middle lobe, right lower lobe, and left lower lobe; (**g**) color-coded T1 OE image showing a large bulla with reduced green foci within in the right upper lobe (white arrow); (**h**) dynamic contrast-enhanced (DCE) perfusion image showing perfusion defects in right upper lobe (white arrow, showing perfusion defect corresponding to the bulla), right middle lobe, right lower lobe, and left lower lobe.

**Figure 6 diagnostics-13-03511-f006:**
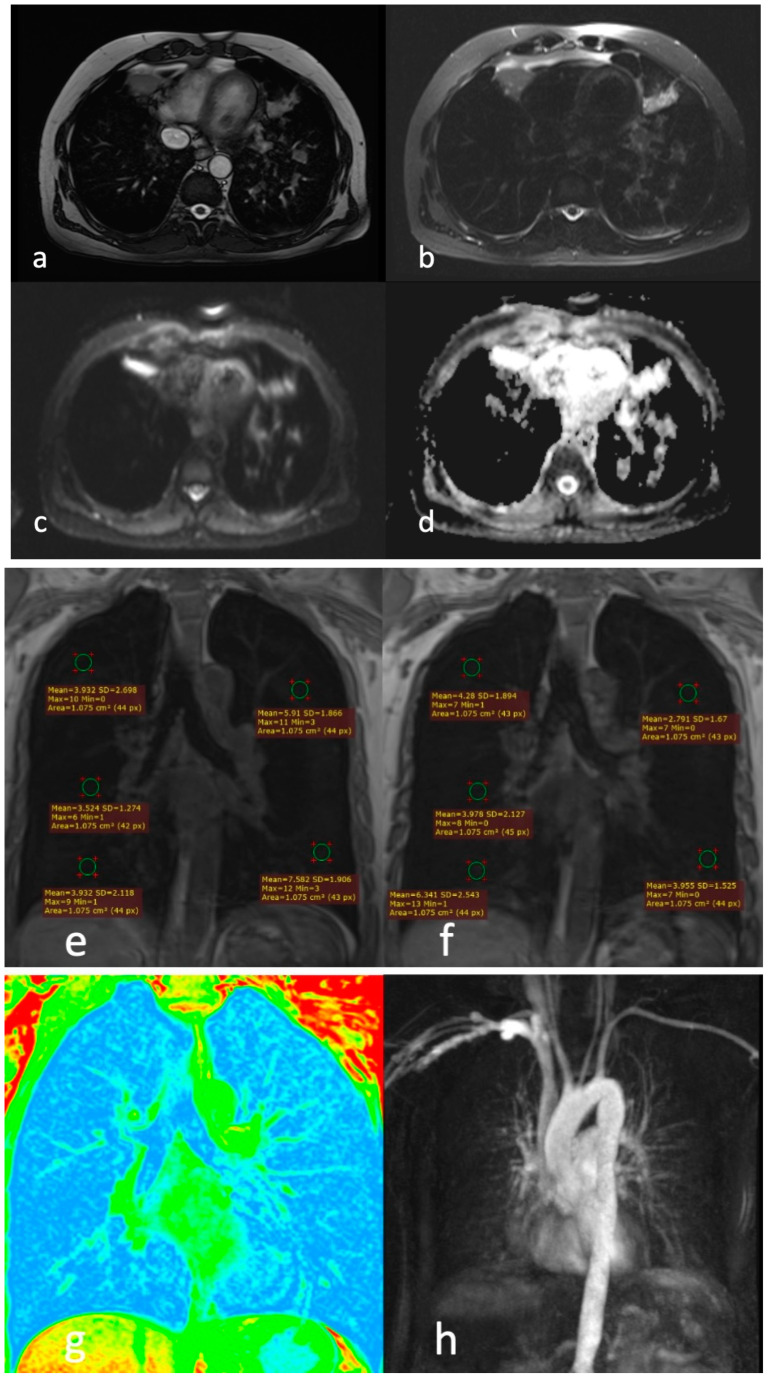
A 54-year-old male patient with GOLD III severity score; (**a**) T2 HASTE, (**b**) T2 TRUFI, and (**c**) T2 BLADE images showing patchy areas of consolidation in right middle lobe, left upper lobe, and left lower lobe (white arrow); (**d**) DWI images showing non-diffusion restricting patchy areas of consolidation in right middle lobe, left upper lobe, and left lower lobe; (**e**) T1 image in room air (RA) and (**f**) T1 image, oxygen enhanced (OE), showing ventilation defects in the right middle lobe, left upper lobe, and left lower lobe; (**g**) color-coded T1 image showing no obvious ventilation defects; (**h**) dynamic contrast-enhanced (DCE) perfusion image showing mild perfusion defects in all lobes.

**Table 1 diagnostics-13-03511-t001:** Table depicting MRI protocol used in the study.

	T2W HASTE	T2W TRUFI	T2W BLADE	DIFFUSION	T1	TWIST ANGIO 3D
Plane	Axial	Axial	Axial	Axial	Coronal	Coronal
TR/TE (ms)	500/39	448/144	2500/118	5100/56	3.57/1.35	2.58/0.98
Slice thickness (mm)	5	5	5	6	15	5
Slice per station (n)	30	56	40	30	28	28
FOV (mm)	320	320	320	400	450	450
Acquired voxel size (mm^3^)	1.3 × 1.3 × 5	0.6 × 0.6 × 5	1.3 × 1.3 × 5	1.3 × 1.3 × 6	1.4 × 1.4 × 15	0.9 × 0.9 × 4.5
Bandwidth/ pixel (Hz)	781	1028	501	2200	390	740
Flip angle	146	70	150		8	25
Acquisition time	30 s	50 s	72 s	150 s	9 s	70 s

**Table 2 diagnostics-13-03511-t002:** Demographics of the study population.

Number of Participants	*n* = 60	
**Sex**		
Male	56	(93.3%)
Female	4	(6.7%)
**Age distribution**		
<50 years	6/60	(10%)
51 to 60 years	16/60	(26.7%)
61 to 70 years	30/60	(50.0%)
>70 years	8/60	(13.3%)
**Symptoms**		
Shortness of breath	58/60	(96.7%)
Cough	38/60	(63.3%)
Expectoration	18/60	(30.0%)
Wheeze	42/60	(70.0%)
**GOLD classification distribution**		
GOLD I	4/60	(6.7%)
GOLD II	20/60	(33.3%)
GOLD III	32/60	(53.3%)
GOLD IV	4/60	(6.7%)
**FEV (% of predicted)**		
>50	24/60	(40%)
<50	36/60	(60%)

(GOLD—Global initiative for chronic Obstructive Lung Disease, FEV1—Forced Expiratory Volume of first second).

**Table 3 diagnostics-13-03511-t003:** Comparison of ventilation and perfusion defects (obtained using a Likert scale) with GOLD classification.

		Average of Ventilation and Perfusion Defects			*p* Value
		Mean	SD	Median	
GOLD Classification score	I	1.00	1.41	1	0.104
	II	4.40	3.06	4	
	III	6.38	3.14	8	
	IV	4.50	3.54	5	

## Data Availability

Data are contained within the article.

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
