# Peer review of "Evaluation of Dynamic Contrast-Enhanced and Oxygen-Enhanced Functional Lung Magnetic Resonance Imaging in Chronic Obstructive Pulmonary Disease Patients"

_diagnostics, 2023, doi:10.3390/diagnostics13233511_

Round 1
Reviewer 1 Report
Comments and Suggestions for Authors
The authors report on use of MRI to detect ventilation and perfusion in the lung- finding some benefits that roughly relate to lung function.
Comments:
The imaging seems similar to what is obtained with a V Q scan ( which have not proved to be very helpful in COPD)
The statement in the final paragraph "It can be used in the assessment of treatment response, planning for surgical volume reduction, and endobronchial valve placement to assess the regional lung status. " is not supported by any data presented and needs to be modified.
Comments on the Quality of English LanguageNeeds some editing for grammar and syntax
Reviewer 2 Report
Comments and Suggestions for Authors
In this manuscript, the authors present their work using oxygen-enhanced and dynamic contrast enhanced MRI to evaluate patients with COPD. This is an exciting study in that the authors have a large sample size with which to work, which is relatively uncommon in the functional lung imaging literature. However, the methods could be more clearly described, and the discussion needs to be streamlined. In addition, several of the figures could be improved to more clearly illustrate the methods and results.
Specific Comments:
1) Page 2, Lines 65-68: The explanation of why MRI is difficult in the lungs could be improved. I do not think the current explanation would be understandable to a non-expert. There is no mention of respiratory or cardiac motion.
2) Page 2, Lines 71-74: Likewise, the explanation of hyperpolarized gases could be made more clear.
3) Page 2, Line 80, and throughout the manuscript: Throughout the manuscript, the authors appear to conflate T1 weighted imaging and T1 mapping. It is not at all clear whether T1 mapping was actually performed or if you just used T1 weighted imaging on room air and with 100% oxygen. The manuscript should be reworked for clarity on this point.
4) Page 3, Line 83: There is a big jump from the discussion of oxygen enhanced MRI to DCE MRI. This should be a new paragraph and give a little additional introduction to the DCE technique.
5) What was the point of imaging 2 controls? That isn't enough to do any statistical analysis, and their data wasn't really used. I think the controls could just be left out of the paper.
6) Methods: The explanation of the collection of images is not terribly clear for OE or DCE MRI. It would help to rework this explanation for clarity and perhaps cite papers that used similar methods.
7) Page 5, Line 135: You say that "good contrast related enhancement " ... " was considered normal". What is "good"? How is this decided?
8) Page 5, Line 139: 18.6 s seems like a really long breath-hold for COPD patients. Some discussion of how well this long breath-hold was tolerated would be useful.
9) The Likert scale rating is an interesting choice - Most OE and DCE literatures uses quantitative measures such as VDP or QDP. It would be helpful to additionally calculate these more common measures to see how they compare to the semi-quantitative measure.
10) Page 5, Lines146 - 148. It would be helpful to include a figure (or add to one of the current figures) an illustration of how the lungs were divided and evaluated lobe-by-lobe.
11) Figure 2: A colorbar should be included. Also, in c, there appears to be artifact obscuring the periphery of the lungs - Some explanation of this would be helpful.
12) Figure 3: I think it would be helpful to separate all four GOLD stages in the box and whisker plot. And to note significant statistical differences between groups with lines or p-values. I'm not sure what the grand median adds.
13) Figures 5 and 6: The labels on the ROIs are distracting - They're really too small to read, so they just obscure the view of the images. Also, there is no mention of the ROIs in the captions, so not sure that they're adding anything.
14) The first paragraph of the discussion section reads more like an introduction (and it largely repeats what was already introduced). This paragraph could be omitted entirely (although I'll note that the description of challenges and successes in pulmonary MRI is better here than in the introduction).
15) On the whole, the discussion seems scattered and jumps from idea to idea. There should be better transition from idea to idea in order to help the reader.
16) Page 12, Line 283: You mention that one of the major drawbacks of OE MRI is low SNR. With this statement, it would make sense to discuss the SNR that you observed in this study - Was this limitation overcome?
17) Page 12, line 304: You mention that the ancillary findings on MRI could suggest a probable cause for the acute exacerbation - This is the first time acute exacerbation was mentioned. In your methods section, it doesn't sound as though you recruited patients with acute exacerbations. It isn't at all clear whether your patient population had exacerbations or not. This should be clarified.
18) There was really no mention of CT imaging until it is introduced in the discussion section (Page 13, Line 320). Such comparison should be added to the results section.
19) The conclusion paragraph contained a lot of speculation about how this could be used in the future with no underlying evidence. This paragraph should be reworked to present the overall findings from the paper and temper any big claims (e.g., you say that it can be used in the assessment of treatment response, which you did not show in this paper).
Comments on the Quality of English Language
Generally, the writing is done well, although the discussion section in particular really jumps from thought to thought with little transition. This should be streamlined and more carefully tied to the results section in order to make it easier for the reader to understand the authors' line of thinking.
Round 2
Reviewer 2 Report
Comments and Suggestions for Authors
The authors have addressed the majority of my comments. I have a few minor remaining critiques:
1) In response to comment 6, the authors point out that their methods were similar to those used in citation 12. To assist the reader, it would be very helpful to have this explicitly noted in the methods section. Something like: "Methods for OE and DCE MRI are similar to those reported in (12).".
2) In response to comment 10, the authors state that the lobes were divided based on fissures. In which image? In my experience, it is often very difficult to visualize fissures on MRI. I can't see any fissures in any of the images shown in the figures for this paper. Additionally, was this done manually or in an automated fashion? Some additional information would be helpful.
3) In response to Comment 11, the authors added a colorbar to Figure 2, which I appreciate. However, the units on the colorbar are in ms. In response to comment 3, the authors stated that they did not perform T1 mapping, but only T1 weighted imaging. As such, it is not clear to me how they would have gotten units of ms for Figure 2? It seems like this should perhaps be arbitrary units instead? Or units of fractional ventilation? In addition, the authors respond that there are indeed artifacts in panel C of figure 2. I totally agree, and I think this should be noted in the figure caption, perhaps with a note on how this is handled in analysis. A fairly sizeable chunk of the lung is omitted, which could impact the Likert scoring.
4) I do not feel as though the authors truly responded to comment 12 - While I understand that they have written the description of their mean measurements in the text and table 3, their figure is unclear - Why were GOLD I/II and GOLD III/IV combined? Why is the grand median shown? As far as I can tell, the p-value comparing the combined groups is never shown. It is not clear what statistical test was used (Methods section says ANOVA and Kruskal Wallis for more than 2 groups, but it doesn't say what is used for 2 groups).
5) In their response, the authors make the argument that the ROIs need to be left in because they demonstrate where the signal intensity was calculated. I would argue that the ROIs themselves should be left in, but the labels should be removed, as they are distracting. It doesn't help the reader, because the labels are too small to read, and thus is just distracting. Ultimately, this is a stylistic point, so it is ultimately up to the authors, but I would encourage them to consider removing labels but leaving in the circles for the ROIs.
6) I believe the authors may have misunderstood my point in Comment 16. The authors make reference to the fact that SNR limitations have hindered OE MRI in the past, but there is really no reference to SNR throughout the manuscript. I would argue that the authors should do one of two things:
* Move this statement to the introduction. If you aren't presenting any information, then there is no reason to have this in the discussion section.
* Add the average SNR of OE MRI to the results section (Or perhaps to Table 1), so that you can actually point to this as evidence that you have in fact overcome this limitation.
Frankly, the entire back half of that paragraph (Lines 274-280) seems out of place and could be omitted entirely in my opinion.
